# Spatio-Temporal Changes of Vegetation Cover and Its Influencing Factors in Northeast China from 2000 to 2021

**Maolin Li** [1,2] , **Qingwu Yan** [1,3,*], **Guie Li** [1,3], **Minghao Yi** [1,3] **and Jie Li** [1,2]

[1] Observation and Research Station of Ministry of Education for Resource Exhausted Mining Area Land Restoration and Ecological Succession, China University of Mining and Technology, Xuzhou 221116, China

[2] College of Environment Science and Spatial Informatics, China University of Mining and Technology, Xuzhou 221116, China

[3] School of Public Policy and Management, China University of Mining and Technology, Xuzhou 221116, China

\* Correspondence: yanqingwu@cumt.edu.cn

**Abstract:** The foundation of study on regional environmental carrying capacity is the detection of vegetation changes. A case of Northeast China, we, with the support of normalized difference vegetation index (NDVI) of MOD13A3 (MOD13A3-NDVI), use a three-dimensional vegetation cover model (3DFVC) to acquire vegetation cover from 2000 to 2021. Vegetation trends are then monitored by the spatio-temporal analysis models including the empirical orthogonal function (EOF), the Sen's slope (Sen), the Mann-Kendall test (MK) and the Hurst index (Hurst). Additionally, we, through the multi-scale geographically weighted regression model (MGWR), explore the spatial heterogeneity of vegetation response to its influencing factors. On the basis of this, it is by introducing the structural equation model (SEM) that we figure out the mechanisms of vegetation response to climate and human activity. The main results are as follows: (1) Compared with the dimidiate pixel model (FVC), 3DFVC, to some extent, weaken the influence of terrain on vegetation cover extraction with a good applicability. (2) From 2000 to 2021, the average annual vegetation cover has a fluctuating upward trend ($0.03 \cdot 22a^{-1}$, $p < 0.05$), and spatially vegetation cover is lower in the west and higher in the east with a strong climatic zoning feature. In general, vegetation cover is relatively stable, only 7.08% of the vegetation area with a trend of significant change. (3) In terms of EOF ($EOF_1+EOF_2$), $EOF_1$ has a strong spatial heterogeneity but $EOF_2$ has a strong temporal heterogeneity. As for the Hurst index, its mean value, with an anti-persistence feature, is 0.451, illustrating that vegetation is at some risk of degradation in future. (4) MGWR is slightly better than GWR. Vegetation growth is more influenced by the climate (precipitation and temperature) or human activity and less by the terrain or soil. Besides, precipitation plays a leading role on vegetation growth, while temperature plays a moderating role on vegetation growth. What is more, precipitation, on different temperature conditions, shows a different effect on vegetation growth.

**Keywords:** vegetation trends; MGWR; spatial heterogeneity; zoning characteristics; factor analysis

## 1. Introduction

In the terrestrial ecosystem, vegetation is both an essential component and a key producer. It links several environmental components, including the atmosphere, soil, and groundwater, and has an impact on the region's biodiversity and ecological quality [1,2]. Vegetation cover is defined as the vertical projection area of the above-ground portion of vegetation (including the leaves, stems and branches) as a percentage of the total surface area [3]. It is not only a vital parameter for portraying the surface vegetation and a basic indicator in the ecological environment, but also occupies a key position in the atmosphere, soil and biosphere [4,5]. Therefore, it has both theoretical and practical significance for the management of ecological process and the protection of ecological environment to monitor spatio-temporal variations in vegetation cover and figure out the driving mechanism of vegetation.

The extraction of vegetation cover is to some extent dependent on the extraction methods. At present, the methods for extracting vegetation cover, by remote sensing, can be mainly classified into three types, including the regression method [6], the mixed-pixel method [7] and the machine learning method [8]. Dimidiate pixel model (FVC) is one of the most representative models in these methods, and is widely used for the estimation of vegetation cover due to its formal simplicity and easy calculation [9]. However, it is when the terrain is relatively complex that FVC has certain drawbacks. For instance, when estimating vegetation cover, FVC usually idealizes the area occupied by vegetation in a pixel as a two-dimension flat, while in reality, the area occupied by vegetation in a pixel is a three-dimension curved surface. Besides, what a certain loss of information may happen to replacing a three-dimensional curved surface with a two-dimensional flat to estimate vegetation cover, which cannot represent the true level of vegetation cover. Therefore, in order to weaken the influence of terrain on vegetation cover extraction, this study, by means of the digital elevation model (Dem), modifies FVC, and the modified model is called three-dimensional vegetation cover model (3DFVC).

In the context of global climate change, many studies on the driving forces of vegetation have been carried out at home and abroad in recent years. For instance, Cheng, using the trend analysis and residual analysis, investigates vegetation trends in Qinling Mountains, and quantifies the relative contributions of human activities and climate to vegetation trends [10]. Otto, with the support of MODIS-NDVI, uses the regression analysis to explore the relationship between vegetation and precipitation in north-western Morocco, and carries out analysis and discussion on vegetation during the growing season [11]. Liu acquires the actual and potential residuals in vegetation changes by means of the residual analysis and has zoning statistics for the results obtained [12]. Besides, there are also many related studies on the driving forces of vegetation, but they remain somewhat inadequate. For example, some studies focus on vegetation response to climate changes [13,14], but ignore the fact that vegetation growth is affected by multiple factors. Therefore, only taking the impact of climate on vegetation growth into consideration may make the conclusions one-side. On the basis of this, this study, from multiple factors (including climate, terrain, soil and human activity) [15–17], explores the driving mechanism of vegetation in combination with the natural and social characteristics in Northeast China. Additionally, in terms of the analysis of affecting factors in vegetation, related studies mainly make use of the correlation or regression analysis to investigate the connections between the affecting factors and vegetation [18,19], and reveal the spatial heterogeneity characteristics in vegetation [20]. However, it is between the affecting factors and vegetation that the relationships are relatively complex, which indicates that there may exist other relationships (except for the correlation or regression) between the affecting factors and vegetation. Structural equation model (SEM) is a method for estimating and testing causality, and it can replace many methods (including the regression analysis, path analysis and other methods, etc.) to diagnose the interrelationship among indicators [21]. As a result, this study uses SEM to figure out the driving mechanism of vegetation, which is of great importance for the restoration and reconstruction of ecology in different regions.

As is known to us, there are few studies on the vegetation cover of the whole of Northeast China, and most studies only cover parts of Northeast China. In this study, it aims to explore the spatio-temporal evolutionary process and spatial heterogeneity in vegetation in Northeast China, and mine the intrinsic driving mechanisms of vegetation growth, which is of great significance for the ecological security governance and sustainable development in Northeast China.

## 2. Study Area and Data Sources

### 2.1. Study Area

With a total area of around $1.44 \times 10^6$ km$^2$, Northeast China (37.95°–53.56°N, 111.15°–135.09°E) is a significant grain production base in China. It is bordered by mountains and rivers and has a large expanse of rich land [22,23]. In terms of climate, Northeast China has a temperate continental climate with cold winters and complex climatic zones. On the one hand, Northeast China has

warm, temperate, and cold temperate variations in temperature from south to north. On the other hand, Northeast China, in precipitation, owns humid, semi- humid, and semi-arid variations from east to west. Moreover, the elevation of Northeast China is high in the west and low in the east, according to its topography map (Figure 1). Northeast China is one of most sensitive areas in ecological environment due to its complex climatic zones and intense human activities [24,25].

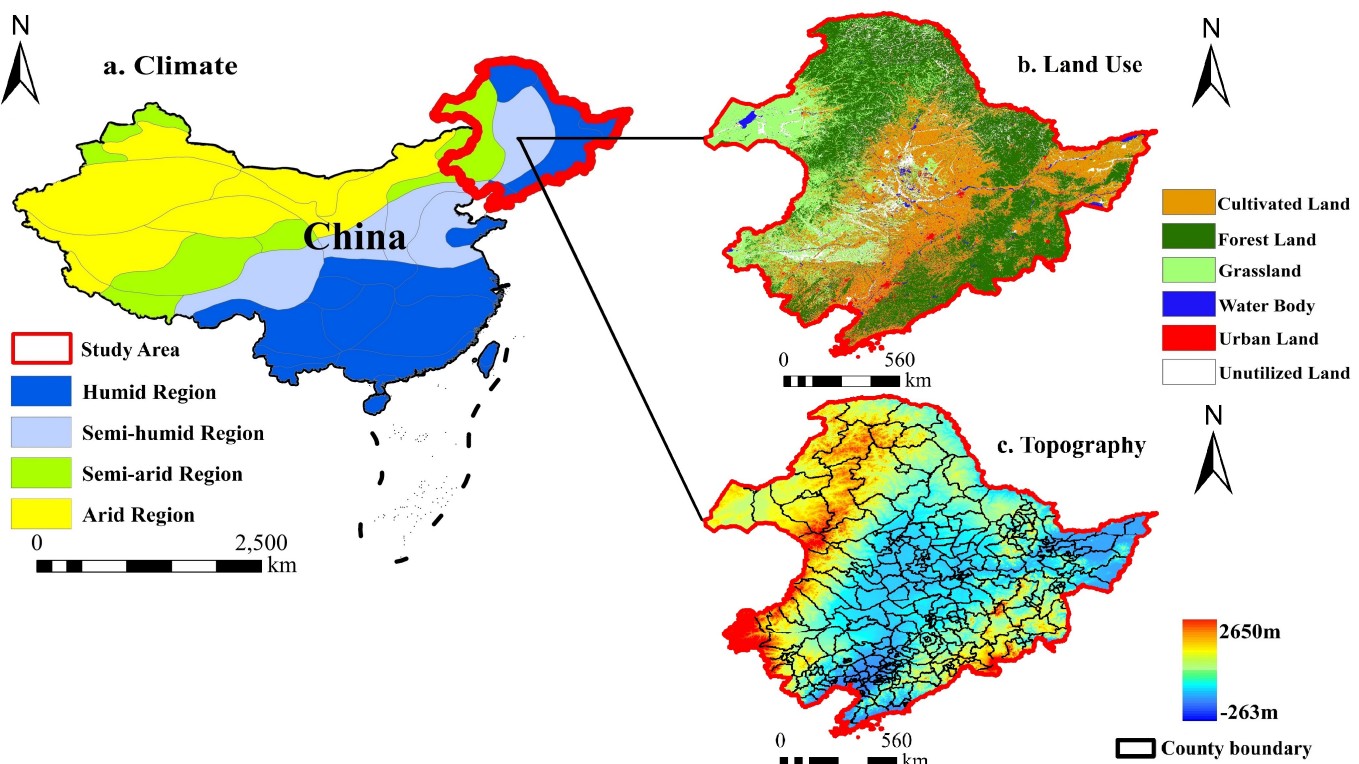

**Figure 1.** Location and climate zones map (**a**), land-use map (**b**) and topography (**c**) of study area. Land-use map, including six categories, was monitored in 2021.

## 2.2. Data Sources

The dataset used in this study ranges from 2000 to 2021 and primarily contains vegetation data, climate data, topography data, soil data and human activity data (Table 1). Vegetation data ($NDVI_{max}$) with a spatial resolution of 1000 m, which is derived from MOD13A3-NDVI, is synthesized by the maximum value compose method (MVC) in a year. Climate data in the meteorological station includes precipitation (Pre) and temperature (Tem), which are processed as Pre and Tem in raster type by Kriging [26]. Topography data originates from SRTM DEM, which is processed as slope with the help of ArcMap 10.6 software. Soil data includes silt, clay and sand, which are expressed as a percentage respectively. Landsat images are derived from the Google Earth Engine (GEE). With the aid of ENVI 5.3 software, we process the Landsat images and acquire land-use classification data (including cultivated land, forest land, grassland, water body, urban land and unutilized land). Moreover, land-use classification, with the help of the comprehensive index of land-use degree method [27], is processed as land-use degree data which is used to represent human activity data (Ha). With the help of ArcMap 10.6 software, all raster data are resampled into 1000 m × 1000 m.

**Table 1.** Data types and data sources.

| Data Type | Resolution | Data Source | Time |
|---|---|---|---|
| Vegetation Data: MOD13A3-NDVI | 1000 m | United States Geological Survey (USGS) https://lpdaac.usgs.gov/accessed on 12 July 2022 | |
| Climate Data: Precipitation; Temperature | null | Meteorological Data Centre of China http://data.cma.cn/accessed on 16 June 2022 | |
| Topography Data: Dem; Slope | 1000 m | Resource and Environment Science and Data Center https://www.resdc.cn/accessed on 18 August 2022 | 2000–2021 |
| Human Activity Data: Landsat | 30 m | Google Earth Engine (GEE) https://code.earthengine.google.com/accessed on 17 June 2022 | |
| Soil Data: Sand; Clay; Silt | 1000 m | Resource and Environment Science and Data Center https://www.resdc.cn/accessed on 15 July 2022 | |
| Boundary Data: Shapefile | null | National Platform for Common Geospatial Information Services https://www.tianditu.gov.cn/accessed on 12 August 2022 | 2021 |

## 3. Methodology

Figure 2 shows the frameworks of study, which includes three steps. The first step is to compare FVC's applicability for complex terrain with 3DFVC's. In this work, we calculate $NDVI_{max}$ into vegetation cover (including $FVC_{1000}$ and $3DFVC_{1000}$ from 2000 to 2021) by means of FVC and 3DFVC. In order to reduce errors, $FVC_{1000}$ and $3DFVC_{1000}$ are processed as multi-year averages respectively. The extraction outcomes of FVC and 3DFVC under different terrain situations are then compared and analyzed. According to their outcomes, 3DFVC with a better applicability is selected for the next study. The second step is to carry out the spatio-temporal analysis in vegetation changes, which is mainly include three subsections. The first subsection is to use Sen and MK to study overall evolutionary trends in vegetation; the second subsection, in different periods or local areas, is to use EOF to analyze the spatio-temporal characteristics in vegetation; and the last subsection is to use Hurst to forecast the future trends in vegetation. What is more, the third step is to investigate the spatial heterogeneity features in vegetation and the intrinsic driving mechanism in vegetation growth. More details are given in the following sections.

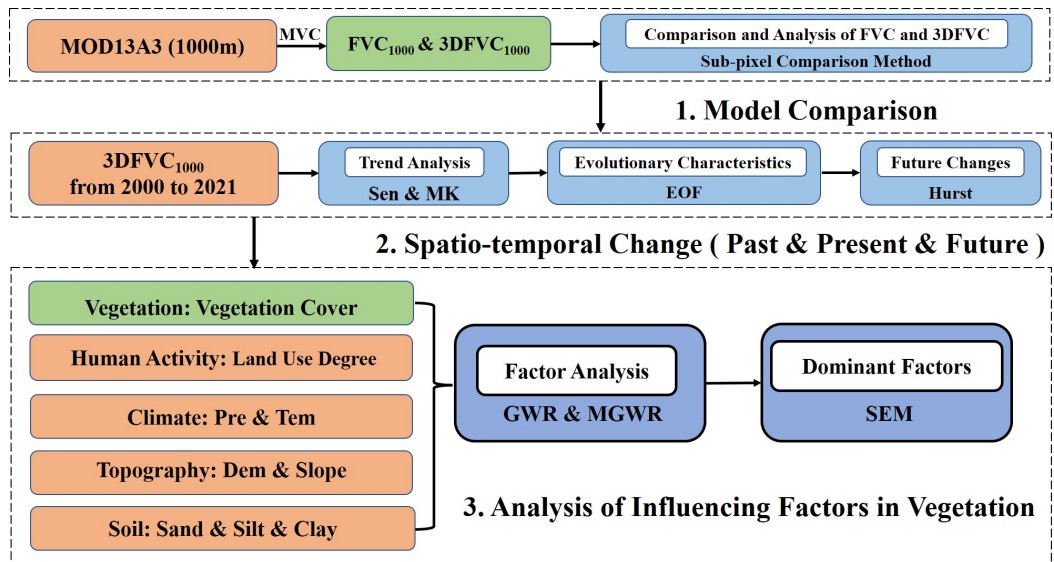

**Figure 2.** Framework of data processing flow.

### 3.1. Vegetation Cover Model

3.1.1. Three-Dimensional Vegetation Cover Model

In this study, the dimidiate pixel model (FVC) is corrected by means of the idea of calculus. Generally, the area of three-dimensional curved surface is difficult to calculate, but is approximatively approached by calculus method (Figure 3). Similarly, this study

applies this idea to remote sensing images, which means that the spatial resolution in remote sensing images is higher and the area occupied by vegetation cover is closer to a two-dimensional flat in a unit. In the meanwhile, the three-dimensional terrain shows less influence on the estimation of vegetation cover and the vegetation information is closer to the definition of vegetation cover in natural conditions.

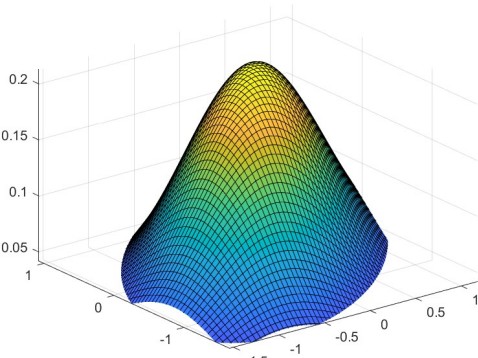

**Figure 3.** Basic principle in calculus method. When the area of curved surface is small enough, it can be approximated as a bevel surface.

According to the definition of vegetation cover, this study regards FVC as the numerator and the three-dimensional curved surface as the denominator, and then acquires a three-dimensional vegetation cover model. The procedures are as follows:

(**1**)　It is assumed that there are two different mixed components (bare land and pure vegetation) in a pixel. Besides, FVC is constructed by the weighted linear combination of two pure components (bare land and pure vegetation) [28]. The equation is as follows:

$$FVC = \frac{NDVI - NDVI_{soil}}{NDVI_{veg} - NDVI_{soil}} \tag{1}$$

where *FVC* represents the dimidiate pixel model, $NDVI_{veg}$ and $NDVI_{soil}$ represents the *NDVI* values of soil and vegetation respectively. In addition, $NDVI_{veg}$ and $NDVI_{soil}$ get their *NDVI* values in the interval of 95% and 5% [29].

(**2**)　Figure 4 shows that *Cosα* is equal to the ratio of $S_{pixel}$ and $S_{slope}$ (the lengths of adjacent and hypotenuse). Similarly, it is obvious that the ratio of $S^2_{pixel}$ and $S_{slope} \times S_{pixel}$ is equal to *Cosα* as well. The equation is as follows:

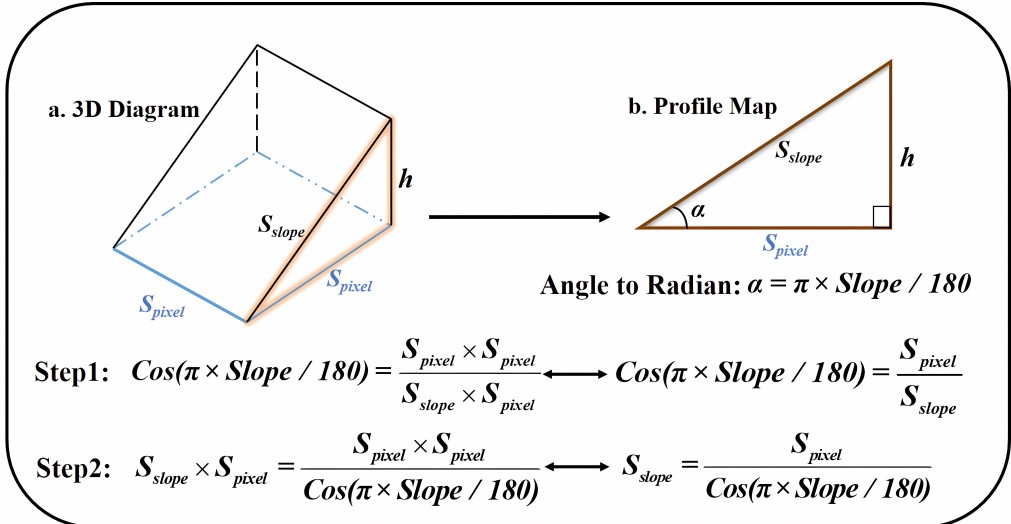

**Figure 4.** The flow of equation derivation for the curved surface.

$$\frac{S_{slope}}{S_{pixel}} = \frac{1}{Cos(\pi \times slope/180)} \tag{2}$$

where $\alpha$ ($\pi \times slope/180$) is the angle between $S_{pixel}$ and $S_{slope}$, and *slope* is originated from Dem and $\pi$ is approximately equal to 3.1415926. $S_{pixel}$ is the side length of the bottom surface and $S^2{}_{pixel}$ is the bottom surface area. $S_{slope}$ is the side length of the bevel surface and the bevel surface area is $S_{pixel} \times S_{slope}$.

(**3**)　This study divides FVC by $1/Cos\alpha$ and gets the three-dimensional vegetation cover model. The equation is as follows:

$$3DFVC = FVC \times Cos(\pi \times slope/180) \tag{3}$$

where *3DFVC* represents the three-dimensional vegetation cover model and *Cos($\alpha$ × slope*/180) indicates the curved surface indicator. In addition, vegetation cover is divided into five categories including high coverage (0.8–1.0), middle high coverage (0.6–0.8), middle coverage (0.4–0.6), middle low coverage (0.2–0.4) and low coverage (0.0–0.2) [30].

### 3.1.2. Accuracy Assessment Method of Vegetation Cover

In this study, sub-pixel comparison method (SCM) is used to evaluate the accuracy of FVC and 3DFVC, and it not only tests the accuracy, but also correlates the obtained vegetation cover with the actual vegetation cover derived from the higher resolution images [31,32]. The images used for the actual vegetation cover are MOD13Q1-NDVI (originated from Google Earth Engine) with a spatial resolution of 250 m, which has the same data processing process as MOD13A3-NDVI. The accuracy of vegetation cover model is higher if the regression coefficient between the vegetation cover to be validated and the actual vegetation cover is closer to 1 [33]. Besides, root mean square error (RMSE) is also used to test the accuracy of vegetation cover model [34]. The equation is as follows:

$$RMSE = \left[ \sum_{i=1}^{n} (X_i - Y_i)^2 / n \right]^{1/2} \tag{4}$$

where $X_i$ indicates the vegetation cover to be validated; $Y_i$ represents the actual vegetation cover; and *n* indicates the samples. Furthermore, the lower the *RMSE* is, the higher the accuracy of vegetation cover model is.

### 3.2. Spatio-Temporal Analysis Model for Vegetation Cover

### 3.2.1. Trend Analysis

Sen's slope (Sen) is a method of trend analysis, which can simulate and explain the trends in vegetation cover from 2000 to 2021 through pixel by pixel [35]. The equation is as follows:

$$\rho = \frac{y_j - y_i}{j - i} (0 < i < j < n) \tag{5}$$

where $y_i$ and $y_j$ represent the vegetation cover of monitoring years *i* and *j*, respectively. Additionally, $\rho$ indicates the trends in vegetation cover. When $\rho > 0$, vegetation coverage has an upward trend; when $\rho < 0$, vegetation coverage has a downward trend; and when $\rho = 0$, vegetation coverage has no change.

The Mann-Kendall trend test (MK) is a method used for significance testing, which removes a small number of outliers and is also applied to the non-standard normal distribution [36]. The equation is as follows:

$$Q = \sum_{i=1}^{n-1} \sum_{j=i+1}^{n} sign(y_j - y_i) \tag{6}$$

$$Z = \begin{cases} \dfrac{Q-1}{\sqrt{\text{Var}(Q)}} & (Q > 0) \\ 0 & (Q = 0) \\ \dfrac{Q+1}{\sqrt{\text{Var}(Q)}} & (Q < 0) \end{cases} \tag{7}$$

where $Q$ is the test statistics; $Z$ is the standard test statistics; and n indicates the samples. When $y_j - y_i > 0$, sign is equal to 1; when $y_j - y_i = 0$, sign is equal to 0; and when $y_j - y_i < 0$, sign is equal to $-1$. When $|Z| > Z_{1-0.05/2} = 1.96$, there is a significant trend in vegetation changes, and when $|Z| \leq 1.96$, there is no significant trend in vegetation changes. Therefore, the study, by overlaying Sen with MK, divides the vegetation trends into five categories (Table 2).

**Table 2.** The categories for vegetation cover trend.

| Theme | Sen | Z | Trend |
|:---:|:---:|:---:|:---:|
| 1 | $\geq 0.0005$ | >1.96 | significant improvement |
| 2 | $\geq 0.0005$ | $-1.96$–1.96 | slight improvement |
| 3 | $-0.0005$–0.0005 | $-1.96$–1.96 | stable |
| 4 | $\leq -0.0005$ | $-1.96$–1.96 | slight degradation |
| 5 | $\leq -0.0005$ | <1.96 | serious degradation |

### 3.2.2. Empirical Orthogonal Function

Empirical Orthogonal Function (EOF) is a statistical method commonly used in the field of atmosphere and ocean [37]. Besides, it primarily, by means of the variance statistics, concentrates the useful information of data on a few spatial distributions and time series so as to reflect the spatio-temporal characteristics in vegetation changes. Assume that there are m observation points in the study area and each observation point has n observation values. The observation matrix is as follows:

$$X = (x_{ij}) = (x_1, x_2, \cdots, x_j) = \begin{pmatrix} x_{11} & \cdots & x_{1n} \\ \vdots & \ddots & \vdots \\ x_{m1} & \cdots & x_{mn} \end{pmatrix} \tag{8}$$

where $x_{ij}$ denotes the j-th observation on the i-th grid. $X$ can be decomposed into a time matrix and a space matrix. The equation is as follows:

$$X = VT \tag{9}$$

where $V$ is the space matrix; and $T$ is the time matrix. $V$ and $T$ are both orthogonal matrices. In addition, $X$ can also be expressed as the product of the spatial eigenvector and the time coefficient. The equation is as follows:

$$x_{ij} = \sum_{k=1}^{m} v_{ik} t_{kj} = v_{i1} t_{1j} + v_{i2} t_{2j} + \cdots + v_{im} t_{mj} \tag{10}$$

where $x_{ik}$ denotes the spatial eigenvector; and $x_{kj}$ indicates the time coefficient. In geography, the spatial eigenvector, called the spatial mode, corresponds to the spatial distributions in study subjects. The larger the variance contribution (VC) of spatial mode is, the better it reflects the spatial distribution characteristics in the eigenvector field. Also, the larger the absolute value of spatial mode is, the greater the magnitude of its change over time is [38]. The time coefficient, which corresponds to the primary component, conveys the time change information of spatial mode in addition to indicating the weight of the spatial mode. The larger the time coefficient is, the more significant the spatial variation characteristics is [39]. In general, EOF can dissect the spatio-temporal development of vegetation by decomposing the temporal and spatial variables.

Moreover, it is necessary to check whether the spatial mode is meaningful before the result is gotten, which is achieved by calculating the error range for the characteristic roots [40]. The equation is as follows:

$$|\lambda_{k+1} - \lambda_k| \geq \lambda_k \times (2/m)^{1/2} \tag{11}$$

where $\lambda_{k+1}$ and $\lambda_k$ represent the eigenvalues, respectively; k indicates the ordinal number in eigenvalues; and m is the total number in eigenvalues. When k = 3, the inequality does not hold ($|14656.16–16205.74| < 16205.74 \times (2/13)^{1/2}$). Therefore, the first and second eigenvalues ($\lambda_1$ and $\lambda_2$) are only reserved. $\lambda_1$ corresponds to the first eigenvector field (EOF$_1$) and its time coefficient; $\lambda_2$ corresponds to the second eigenvector field (EOF$_2$) and its time coefficient; and the variance contributions of $\lambda_1$ and $\lambda_2$ are respectively $VC_1$ and $VC_2$.

### 3.2.3. Hurst Index

Hurst index (Hurst) is a method for quantitatively describing the information dependence [41], which can quantify the relationship between the future trend and the previous trend and try to figure out whether vegetation changes have continuity through time series. In this study, rescaled range analysis method (R/S) is used to reveal the evolutionary characteristics in vegetation future changes [42]. The equation is as follows:

$$X(t, \tau) = \sum_{u=1}^{t} \left( \xi(u) - \frac{1}{\tau} \sum_{t=1}^{\tau} \xi(t) \right) \tag{12}$$

$$R(\tau) = \max_{1 \leqslant t \leqslant \tau} X(t, \tau) - \min_{1 \leqslant t \leqslant \tau} X(t, \tau) \tag{13}$$

$$S(\tau) = \left[ \frac{1}{\tau} \sum_{t=1}^{\tau} (\xi(t) - \xi_\tau)^2 \right]^{1/2} \tag{14}$$

where $X(t, \tau)$ represents the sequences in cumulative deviations ($\tau$=1, 2, 3,$\cdots$, n); $R(\tau)$ indicates the range sequences; and $S(\tau)$ represents the standard deviation sequences. When $R(\tau)/S(\tau)\alpha\tau H$, vegetation cover has a certain continuity in trends. Hurst usually ranges from 0 to 1. When $H$ = 0.5, vegetation cover has no significant change in future; when $H > 0.5$, vegetation cover has a positive continuous trend with a persistence feature in future; and when $H < 0.5$, vegetation cover has a negative continuous trend with an anti-persistence feature in future. In addition, the study overlays Sen with Hurst index, which can divide the vegetation future trends into four categories including continuous improvement (*Sen* > 0 and $H > 0.5$), continuous degradation (*Sen* < 0 and $H > 0.5$), degradation to improvement (*Sen* < 0 and $H < 0.5$) and improvement to degradation (*Sen* > 0 and $H < 0.5$).

### 3.3. Influencing Factor Analysis Model for Vegetation Cover
3.3.1. Multi-Scale Geographically Weighted Regression

Geographical weighted regression (GWR) is an extension for ordinary least squares (OLS), and its weight is a function of distance between the geospatial location of observation point and the regression point, aiming to weigh the influence degree to which the observations at different geographic locations have an impact on the parameter estimates at regression points [43]. Multi-scale geographically weighted regression (MGWR) is an improvement of GWR, which produces multiple adaptive bandwidths at different spatial scales with a good level of spatial smoothing [44]. The equation is as follows:

$$y_i = \sum_{j=1}^{k} \beta_{bwj}(u_i, v_i) x_{ij} + \varepsilon_i \tag{15}$$

where $i$, $j$ represent the sample size and the independent variable size, respectively; $k$ denotes the maximum value of $j$; $y_i$ indicates the explained variable; ($u_i,v_i$) represents the

spatial location; $x_{ij}$ is on behalf of the explanatory variable; $\varepsilon_i$ indicates the stochastic error term; *bwj* represents the bandwidth.

In this study, corrected akaike information criterion (AICc) and R-square ($R^2$) are used to evaluate the goodness in models, which specifically means that a lower AICc indicates a better model superiority and a higher $R^2$ represents a better model fit [45]. In addition, the multicollinearity problem in multiple variables tends to affect the accuracy of regression results. Therefore, we use variance inflation factor (VIF) to test the multicollinearity in multiple variables [46]. When VIF < 10, it means a weak collinearity in multiple variables; when $10 \leq VIF < 100$, it means a strong collinearity in multiple variables; and when $VIF \geq 100$, it indicates a severe collinearity in multiple variables.

### 3.3.2. Mediating Effect Model and Moderating Effect Model

Traditional geological models focus on exploring the direct relationship between vegetation and its influencing factors, but ignore the indirect effects between them in practical applications [47,48]. Structural equation model (SEM) is a statistical method for testing causality [49], which can replace many methods (including the multiple linear regression, path analysis and other methods) to analyze the relationships between each variable and be a multivariate statistical technique used for modeling [50]. In this study, mediating effect model and moderating effect model in SEM are used to explore the potential relationships between them. The rationale is as follows (Figure 5). *M* is the independent variable; *N* is the dependent variable; and *Z* is the mediator or the moderator. Besides, the two models cannot hold simultaneously. As shown in Figure 5a, if *M* has an effect on *N* through *Z*, ME can hold. For example, precipitation has an effect on vegetation and temperature also has an effect on vegetation. However, precipitation does not affect vegetation directly but indirectly through temperature. As shown in Figure 5b, when *Z* has an effect on the relationship between *M* and *N*, MO holds. For instance, precipitation has an effect on the relationship between temperature and vegetation. But when temperature is higher or lower, precipitation has a different effect on vegetation. Moreover, the growth of $R^2$ ($\Delta R^2$) is used to evaluate the models in this study. The higher $R^2$ and $\Delta R^2$ are, the better the fitting results for ME and MO are. For the testing of significance in models, the lower $F_p$ (the significance of F value) and $p$ (the significance of interaction) are, the more significant ME and MO are.

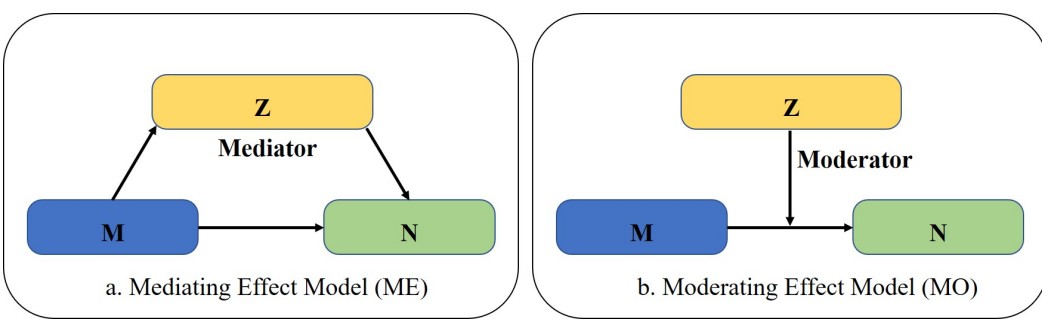

**Figure 5.** Mediating effect model map (**a**) and moderating effect model map (**b**).

## 4. Results

### *4.1. Comparison and Analysis in FVC and 3DFVC*

#### 4.1.1. Comparison of FVC and 3DFVC

In order to reduce random errors, $FVC_{1000}$ and $3DFVC_{1000}$ from 2000 to 2021 are calculated into average annual vegetation cover (including $FVC_{Ave1000}$ and $3DFVC_{Ave1000}$) in this study. The difference ($D_{1000} = FVC_{Ave1000} - 3DFVC_{Ave1000}$) between $FVC_{Ave1000}$ and $3DFVC_{Ave1000}$ is then acquired by subtraction (Figure 6). In general, the more complex terrain is, the higher $D_{1000}$ is. On particular, $D_{1000}$ is higher in Daxing'an Mountains and Changbai Mountains, but lower in the Northeast Plain. It indicates that the difference in FVC and 3DFVC is not significant where the terrain is relatively simple in some areas (such

as plain and platform). However, as the terrain becomes more complex, the difference in FVC and 3DFVC becomes more obvious.

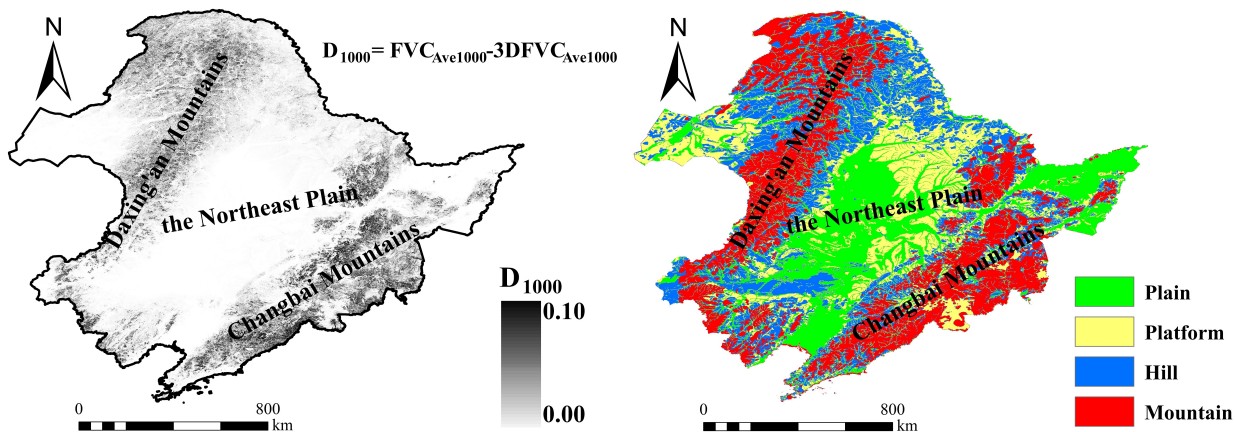

**Figure 6.** The spatial distribution of terrain and $D_{1000}$.

### 4.1.2. Statistical Validation in FVC and 3DFVC

$D_{1000}$ only indicates the difference between FVC and 3DFVC, but does not represent the extraction accuracy of FVC and 3DFVC. Thus, we use sub-pixel comparison method to validate the extraction accuracy of FVC and 3DFVC (Figure 7). The results show that 3DFVC's regression coefficient increases by 0.07 and is closer to 1 than FVC's. But the improvement of 3DFVC's regression coefficient is not extremely obvious, which may be caused by small percentage of complex terrain area in total area, and more details are explained in the terrain analysis. In addition, 3DFVC's root mean square error (RMSE) decreases by 16.5 and is lower than FVC's. Obviously, in contrast to FVC, 3DFVC, with the Cos$\alpha$ function, lowers the influence of complex terrain on vegetation extraction, making itself have a higher extraction accuracy and a stronger applicability.

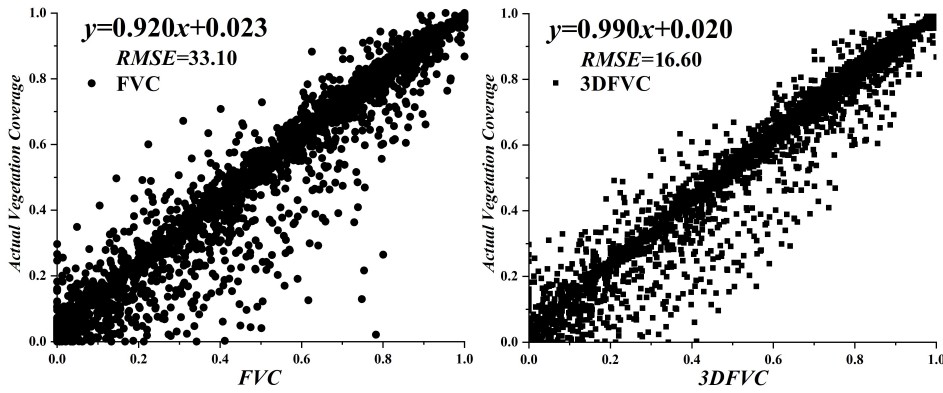

**Figure 7.** Accuracy comparison between FVC and 3DFVC.

### 4.2. Spatio-Temporal Analysis in Vegetation Changes

#### 4.2.1. Spatio-Temporal Characteristics for Vegetation Changes

As shown in Figure 8, vegetation cover is overall high in Northeast China and its mean value is equal to 0.7174. Additionally, spatially vegetation cover has a distinct zonal characteristic (Region I, Region II and Region III). It is necessary to remind that Region II also includes a small piece of region (II) in the southwest.

As shown in Figure 8a, vegetation cover has a fluctuating upward trend from 2000 to 2021 ($k = 1.4 \times 10^{-3}$), ranging from 0.686 to 0.739, with the lowest and highest values occurring in 2001 and 2016, respectively. In addition, the span in vegetation changes can be broadly divided into three periods from 2000 to 2021 ($T_1$ in 2000–2007, $T_2$ in 2007–2013 and $T_3$ in 2013–2021). Vegetation cover increases fast in $T_1$ ($k = 3.9 \times 10^{-3}$), modestly in

$T_2$ ($k = 1.1 \times 10^{-3}$) and slowly in $T_3$ ($k = 0.4 \times 10^{-3}$). Overall, vegetation cover increases significantly in Northeast China from 2000 to 2021, but vegetation cover is gradually stable with passage of time.

As shown in Figure 8b, vegetation cover, with a strong spatial heterogeneity, is high in the east and low in the west, spatially. In particular, the spatial distribution of vegetation cover is correlated with the climate zones to some extent. Region I is in the humid region with a high vegetation cover, Region II is in the semi-humid region with a middle vegetation cover, and Region III is in the semi-arid region with a low vegetation cover. In addition, the spatial distribution of vegetation cover not only associates with the climatic zones, but also has a certain correlation with topography and land use (Figure 1). Therefore, vegetation cover, with a strong spatial heterogeneity, may be influenced by climate, topography or other factors, as detailed in the following sections.

As shown in Figure 8c, vegetation cover in Northeast China is dominated by high and middle high coverage, accounting for more than 70% of the total area. Less than 30% of vegetation area is covered by the middle coverage and below. In addition, from 2000 to 2021, the area of high coverage increases significantly, the area of middle high coverage remains relatively stable and the area of middle coverage and below decreases significantly. Overall, vegetation cover gradually evolves from low coverage to high coverage in Northeast China.

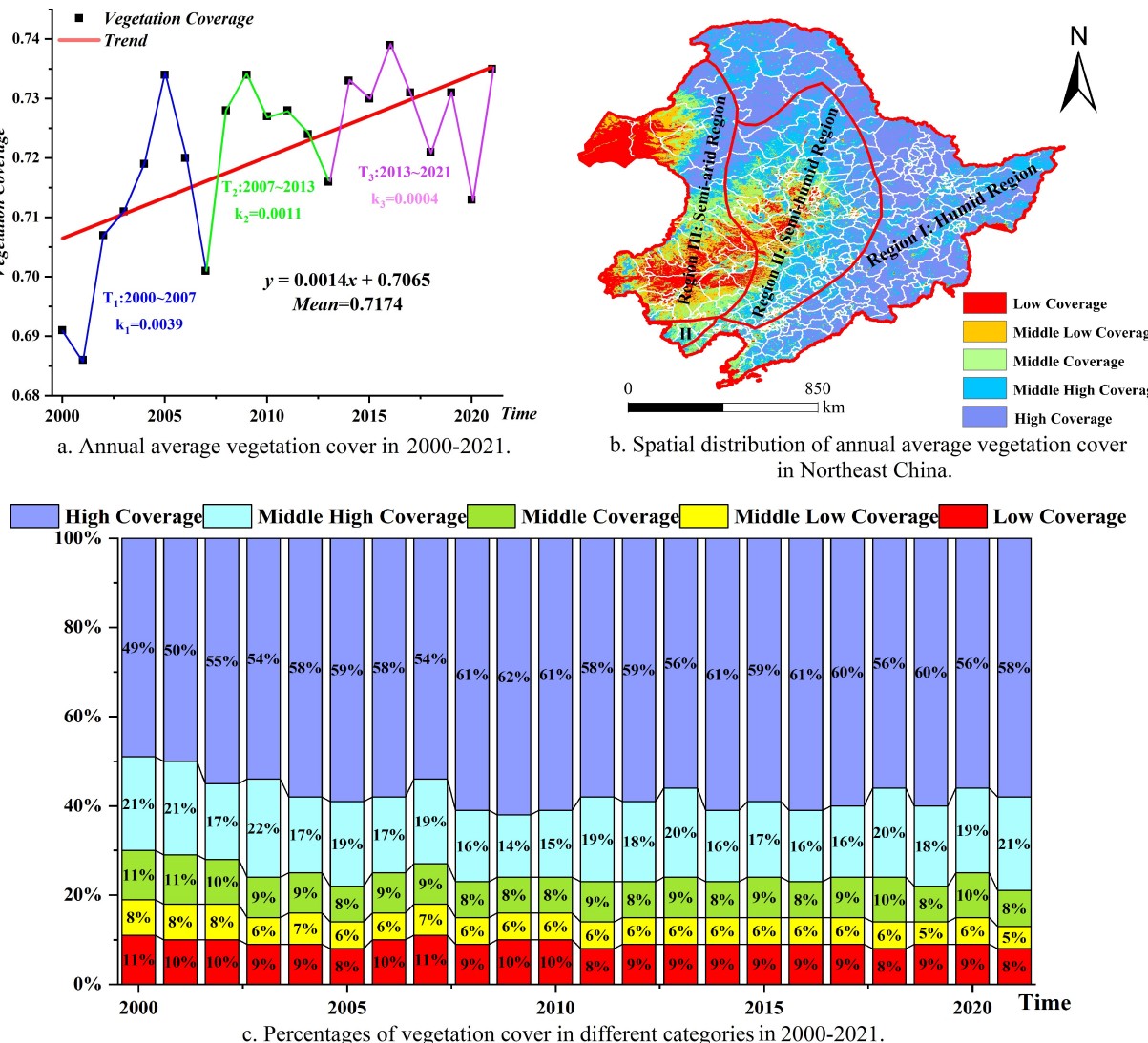

a. Annual average vegetation cover in 2000-2021.

b. Spatial distribution of annual average vegetation cover in Northeast China.

c. Percentages of vegetation cover in different categories in 2000-2021.

**Figure 8.** Spatio-temporal characteristics in vegetation changes.

### 4.2.2. Spatio-Temporal Evolution in Vegetation Cover

As shown in Figure 9, we use Sen and MK to acquire the spatio-temporal trends in vegetation change. The results show that the vegetation trends in most areas are relatively stable. Besides, 60.32 % of vegetation area shows a stable trend, 24.36% of vegetation area shows an improving trend and 15.32% of vegetation area shows a degrading trend. It is in terms of 24.36% of vegetation area with an improving trend that 19.08% of vegetation area has a slight improvement and only 5.28% of vegetation area has a significant trend of improvement. The regions with a significant improvement of vegetation are mostly found in Regions II and Region III, which is probably due to the fact that vegetation cover in these areas is previously low and is significantly improved with the implementation of Grain for Green Project later on. Only 1.80% in 15.32% of vegetation area with a trend of degradation shows a significant trend of degradation, whereas 13.52% of vegetation area shows a slight trend of degradation. What is more, the areas with a significant vegetation degradation are mainly found in the north of Region III, which is likely a result of the region's semi-arid climate and lack of heat in high latitude locations. The vegetation in Region I shows a relative stable trend overall, but changes significantly in local areas. Additionally, especially in the east of Region I, vegetation has a significant trend, which is probably caused by the dual impact of human activity, such as ecological governance and resource development.

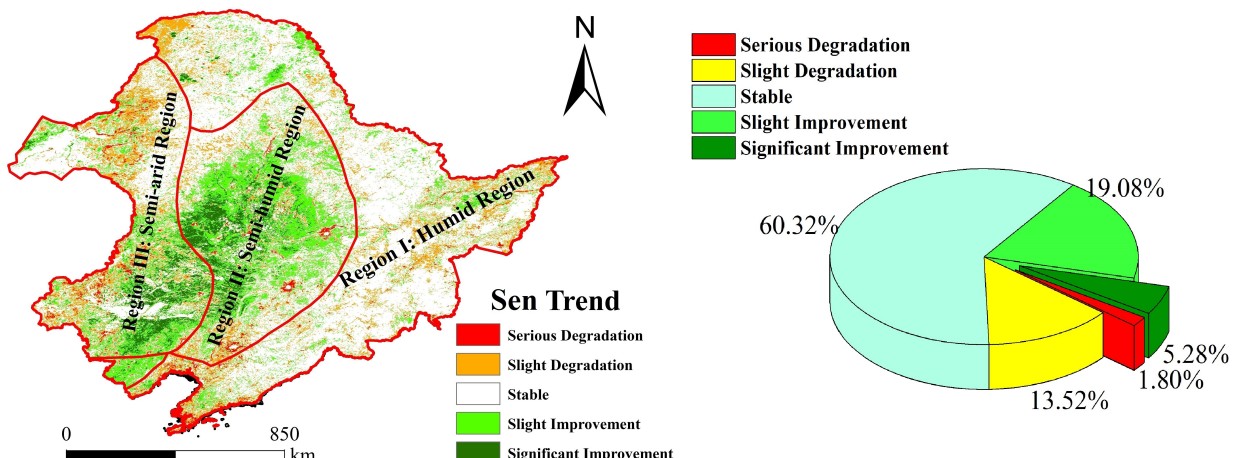

**Figure 9.** Distribution of vegetation cover change trends in Northeast China from 2001 to 2021 and its percentages.

Sen and MK can reveal the overall process in vegetation change from 2001 to 2021, but it is in different periods or locations that they never reflect the detailed characteristics in vegetation change. On the basis of this, EOF, in order to uncover more details in vegetation change, is used to conduct the spatio-temporal decomposition in vegetation from 2001 to 2021 in this study, as shown in Figure 10 (including the variance contributions ($VC_1$ and $VC_2$), the eigenvector fields ($EOF_1$ and $EOF_2$) and the time coefficients).

$EOF_1$'s variance contribution is 25.5% and it represents the main spatial distribution features of vegetation in Northeast China. In the first eigenvector field, $EOF_1$ has a good spatial consistency with mostly positive values, and the high values mainly are located in Region II and the south of Region III. According to $EOF_1$'s time coefficient, which has an upward trend ($k = 6.464$), it indicates that vegetation cover overall shows an upward trend from 2000 to 2021. Additionally, vegetation cover obviously increases, particularly in Region II and the south of Region III, which is largely consistent with Sen's results.

$EOF_2$'s variance contribution is 14.2% and it indicates the local spatial distribution features of vegetation in Northeast China. In the second eigenvector field, $EOF_2$'s time coefficient showing an increasing followed by decreasing trend ($k = -1.062$) is different from $EOF_1$'s. Spatially, the values of $EOF_2$ below 0 ($EOF_2 < 0$) are mostly found in the north of Region II and Region III, whilst the values of $EOF_2$ above 0 ($EOF_2 > 0$) are largely

situated in the south of Region II and Region III. Besides, the span of $EOF_2$'s time coefficient can be broadly divided into two periods ($T_1$ and $T_2$). It is during $T_1$ that vegetation is improved in Region II and Region III, but it is during $T_2$ that vegetation is degraded in Region II and Region III. It is obvious that vegetation in Region II and Region III changes sensitively, and vegetation is locally degraded in spite of the overall improvement.

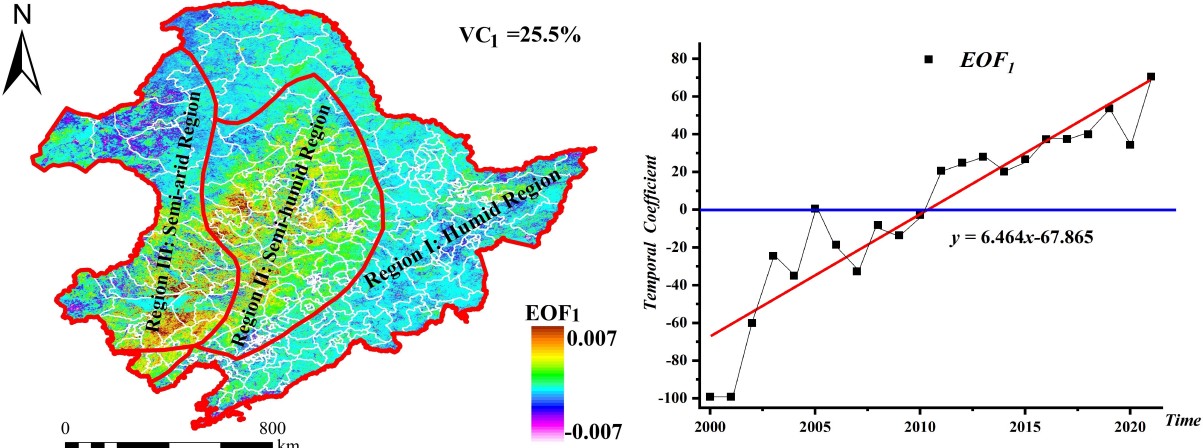

a. The first eigenvector field and its time coefficient.

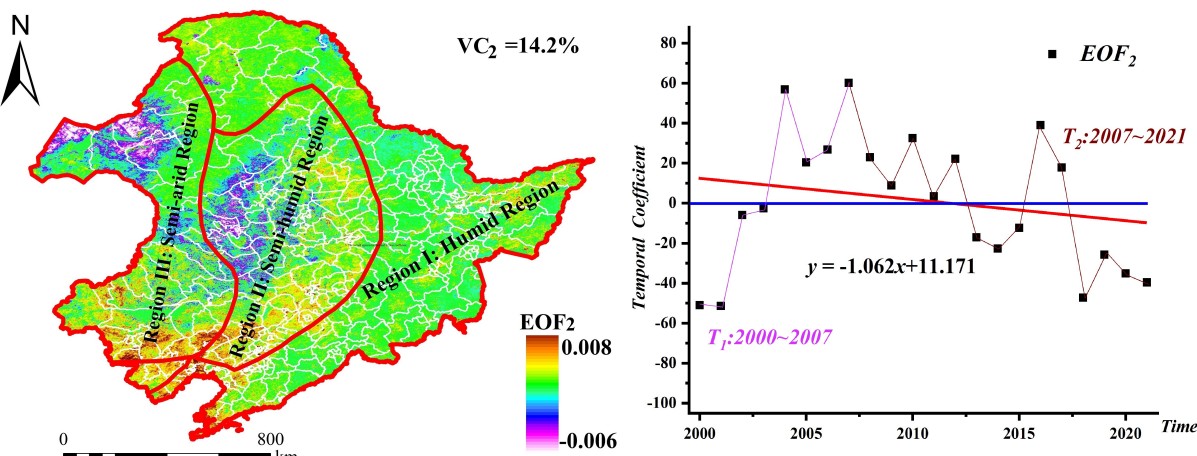

b. The second eigenvector field and its time coefficient.

**Figure 10.** The eigenvector fields and their time coefficients.

Sen, MK and EOF aim to study the spatio-temporal evolution in vegetation, whilst future trend of vegetation is not yet clear. As shown in Figure 11, future trend of vegetation is acquired by overlaying Hurst and Sen in this study.

Hurst ranges from 0.064 to 0.985. Besides, the average Hurst is 0.451 and 70.71% of vegetation area has a Hurst of less than 0.5, which indicates that the anti-persistence feature of vegetation sequence is stronger than the persistence feature of vegetation sequence. The statistics show that 37.56% of vegetation area varies from improvement to degradation, 13.87% of vegetation area has a continuous degradation trend and only 15.42% of vegetation area has a continuous improvement trend. Furthermore, the vegetative area with a Hurst between 0.4 and 0.6 accounts for 64.46% of the total. However, the vegetation area with a Hurst less than 0.4 and more than 0.6 accounts for 29.39% and 6.15% of the total area, respectively, it indicates that local vegetation area has a significant anti-persistence feature in future. It is worth noticing that the Hurst in Region S is higher, which indicates that vegetation change in Region S is going to be more sensitive than in other areas.

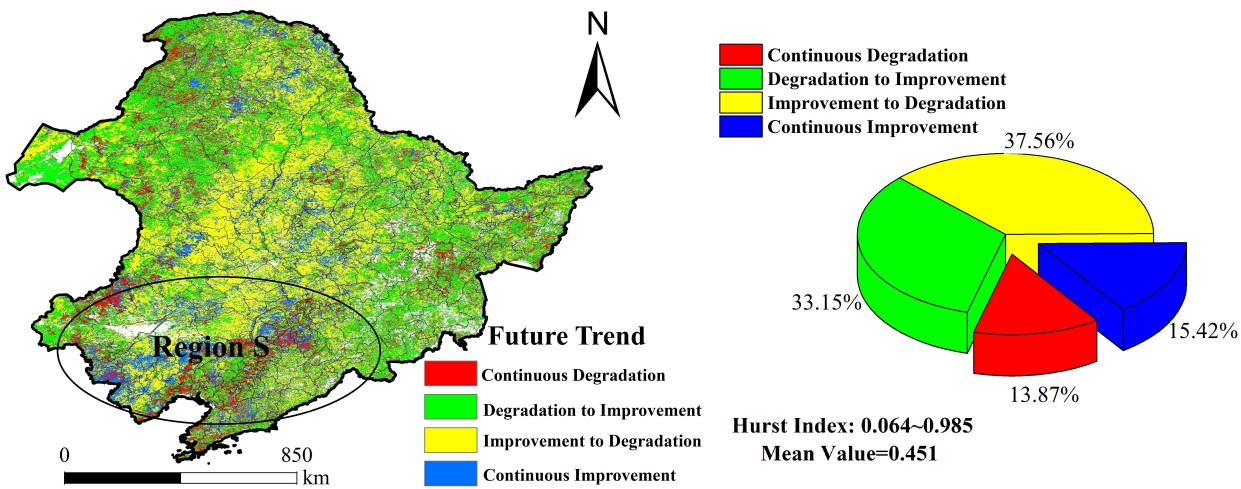

**Figure 11.** Distribution of hurst in Northeast China and its percentages.

### 4.3. Analysis of Influencing Factors in Vegetation Cover

#### 4.3.1. Analysis of Spatial Heterogeneity in Vegetation Changes

The results of spatio-temporal analysis in vegetation show that vegetation in Northeast China has a strong spatial heterogeneity. However, the severe multicollinearity exists between explanatory variables can weaken the model's ability to explain spatial heterogeneity. Therefore, before that, the explanatory variables ought to be tested for multicollinearity, leaving those with a weak multicollinearity. GWR and MGWR then are applied to acquire the regression coefficients for the explanatory variables. Besides, the statistics is shown in Table 3 (including mean, standard error (STD), minimum (Min), maximum (Max), significance ($p$) and VIF).

The results show that the explanatory variables have a weak multicollinearity (VIF < 5) and a high level of significance ($p < 0.05$). In GWR and MGWR, Pre and Clay have a positive influence on vegetation changes (Pre+ and Clay+); Tem, Ha and Dem have a negative influence on vegetation changes (Tem−, Ha− and Dem−); and Slope and Silt have a weak influence on vegetation changes without positive or negative influences significantly.

**Table 3.** Comparison of parameter estimate and testing result for GWR and MGWR. The explanatory variables including precipitation (Pre), temperature (Tem), elevation (Dem), gradient (Slope) and human activity (Ha).

| Model | Variable | VIF | Mean | STD | Min | Max | $p$ |
|---|---|---|---|---|---|---|---|
| GWR | Intercept | | 0.649 | 0.524 | −0.95 | 2.845 | 1.000 |
| | Pre | 2.285 | 0.154 | 0.594 | −0.729 | 1.916 | 0.000 |
| | Tem | 2.15 | −0.236 | 0.856 | −3.772 | 2.406 | 0.000 |
| | Ha | 2.612 | −0.598 | 0.484 | −1.158 | 0.656 | 0.000 |
| | Dem | 3.028 | −0.05 | 0.907 | −3.277 | 3.095 | 0.000 |
| | Slope | 3.985 | 0.037 | 0.472 | −1.004 | 2.924 | 0.000 |
| | Clay | 2.676 | 0.105 | 0.334 | −0.839 | 0.729 | 0.000 |
| | Silt | 1.753 | −0.031 | 0.23 | −0.509 | 0.548 | 0.005 |
| MGWR | Intercept | | 0.617 | 0.005 | 0.607 | 0.622 | 1.000 |
| | Pre | 2.285 | 0.201 | 0.473 | −0.977 | 1.338 | 0.000 |
| | Tem | 2.15 | −0.35 | 0.378 | −0.759 | 0.138 | 0.000 |
| | Ha | 2.612 | −0.706 | 0.459 | −1.324 | 0.777 | 0.000 |
| | Dem | 3.028 | −0.286 | 0.351 | −0.962 | 0.303 | 0.000 |
| | Slope | 3.985 | −0.05 | 0.005 | −0.057 | −0.038 | 0.000 |
| | Clay | 2.676 | 0.128 | 0.264 | −0.878 | 0.622 | 0.000 |
| | Silt | 1.753 | 0.036 | 0.034 | −0.032 | 0.079 | 0.005 |

As shown in Figure 12, it is found that the influencing factors for vegetation cover vary with spatial locations, and there exists a positive or negative correlation between its influencing factors and vegetation growth. The results show that the spatial distribution for influencing factors between GWR and MGWR overall has a high similarity, but there are some differences in local areas. Climate and Ha are the main influencing factors on vegetation growth, with a significant zoning characteristic. On the one hand, it is in GWR that vegetation changes in Region II are primarily influenced by Pre but in Region I are mainly influenced by Ha, with a total of 71.2% of vegetation area influenced by Pre and Ha. On the other hand, it is in MGWR that vegetation changes in Region II are mainly influenced by a combination of Pre and Tem but in Region I are influenced by Ha, with a total of 95.8% of vegetation area influenced by Pre, Tem and Ha.

Vegetation growth is primarily affected by climate in Region II. In terms of Pre, the climatic zones in Northeast China transition from the humid region in the east to the semi-arid region in the west. It is with weak conservation of soil and water that vegetation cover is low in Region II, and vegetation grows mainly from Pre. Therefore, Pre shows a positive dominant influence on vegetation growth in Region II. In terms of Tem, the climatic zones in Northeast China transition from the cold temperate zone in the north to the temperate zone in the south. The heat of cold temperate zone is scarce and vegetation growth is largely dependent on Pre. Besides, the heat of temperate zone is relatively adequate in contrast to the cold temperate zone, which creates favorable conditions for vegetation growth. However, the south of Region II is mainly located in the semi-arid region where the transpiration of vegetation is significant in hot and dry summers. Therefore, Tem shows a negative dominant influence on vegetation growth in the south of Region II.

Vegetation growth is not only affected by the hydrothermal conditions but also susceptible to Ha. In terms of Ha, there is a high land-use intensity in Region I with a relatively suitable climate. It is analyzed that a suitable climate only guarantees the conditions for vegetation growth, but does not play a major role. Especially in the areas of land-use intensity, climate has a weaker impact on vegetation growth than Ha. Therefore, Ha shows a negative dominant effect on the vegetation growth in Region I. Additionally, topography has a less dominant effect on vegetation growth, which has a significant difference in the spatial distribution between GWR and MGWR. It is analyzed that different bandwidth conditions are the reasons why topography has a different spatial distribution in GWR and MGWR. At last, soil also affects vegetation growth, but it is not represented in GWR and MGWR because its estimated parameters are not significant compared to other factors.

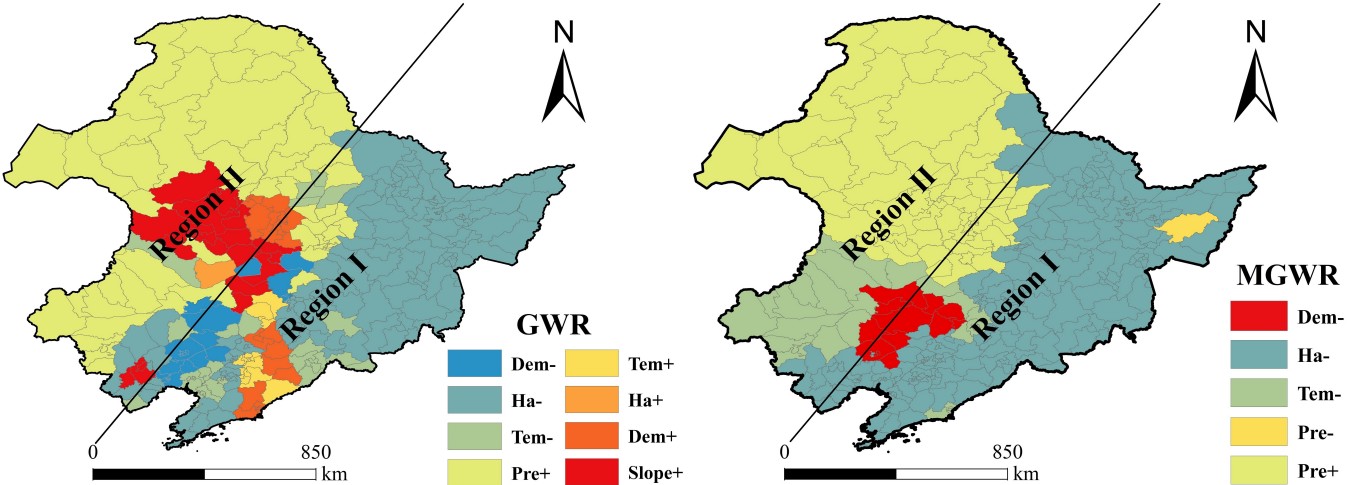

**Figure 12.** Spatial heterogeneity analysis of influencing factors in vegetation. A positive influence on vegetation growth (+); a negative influence on vegetation growth (−) and the influencing factors including precipitation (Pre), temperature (Tem), elevation (Dem), gradient (Slope) and human activity (Ha).

As shown in Table 4, MGWR produces multiple adaptive bandwidths with a good spatial smoothing level. The AICc, BIC and residual sum of squares (RSS) in MGWR are obviously lower than in GWR, which indicates that MGWR uses fewer parameters to make the regression results close to the true values and shows a better explanatory ability. However, it is in terms of adjusted $R^2$ that MGWR's adjusted $R^2$ is only slightly higher than GWR's. In addition, what a quite similar spatial distribution of GWR and MGWR also proves that the differences between MGWR and GWR are not very significant. In conclusion, MGWR is slightly better than GWR in this study.

**Table 4.** Comparison of model superiority in GWR and MGWR.

| Model | Bandwidth | RSS | AICc | BIC | Adjusted $R^2$ |
|---|---|---|---|---|---|
| GWR | 55 | 27.219 | 356.079 | 629.651 | 0.889 |
| MGWR | 27-333 | 25.342 | 268.471 | 499.671 | 0.904 |

4.3.2. Analysis of Dominant Factors in Vegetation Changes

The results of GWR and MGWR indicate that Pre, Tem and Ha play a major role in vegetation growth. As shown in Figure 13, we, in order to figure out the driving mechanisms in vegetation growth, use SEM to further mine the relationships between vegetation and its dominant factors. Besides, the statistics is shown in Table 5 (including the growth value of $R^2$ ($\Delta R^2$), adjusted $R^2$, $F_p$ and *p*).

The results show that there is a significant moderating effect between Pre, Ha and vegetation. Of the three models, model 1 has the highest significance level, adjusted $R^2$ and model explanatory ability, indicating that Pre has a direct effect on vegetation growth and Tem has a moderating effect on vegetation growth. In model 2, when Ha is the moderating variable, $R^2$ is 0.721 with a high significance, indicating that Pre plays a major role in vegetation growth and Ha has a moderating effect on vegetation growth. However, by comparison, it is found that $R^2$ (0.827) for model 1 is higher than that (0.721) for model 2, which indicates that Tem shows a stronger moderating ability over vegetation growth than Ha. In addition, it is when Tem is the independent variable that $\Delta R^2$ for model 3 is 0.343, which indicates that Ha has a certain moderating effect on vegetation growth. However, as a result of the lower $R^2$ (0.349), model 3 has a poor explanatory ability.

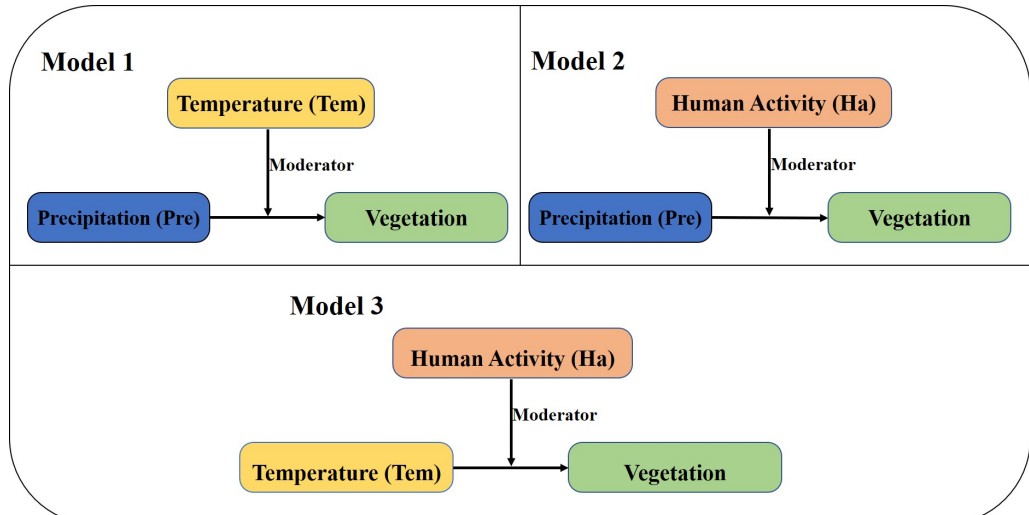

**Figure 13.** Mechanisms of dominant factors on vegetation growth in Northeast China. The dominant factors including precipitation (Pre), temperature (Tem) and human activity (Ha).

**Table 5.** Comparison of model testing result.

| Model | Adjusted R$^2$ | $\Delta$R$^2$ | Fp | *p* |
|---|---|---|---|---|
| 1 | 0.827 | 0.161 | 0.000 | 0.000 |
| 2 | 0.721 | 0.051 | 0.000 | 0.008 |
| 3 | 0.349 | 0.343 | 0.336 | 0.000 |

## 5. Discussion

### 5.1. Strength and Weakness for FVC and 3DFVC

The complex terrain can have a certain effect on vegetation cover extraction, and the most effective way is to conduct a terrain correction on vegetation information [51,52]. 3DFVC, to some extent, weakens the limitation of FVC by introducing Cos$\alpha$, and it is not only suitable for the flat terrain, but also for the complex terrain. It is in terms of the operability in 3DFVC that Cos$\alpha$ can be acquired directly by Dem and its calculation process is extremely simple and quick. 3DFVC is expressed as the ratio of FVC to the curved surface area, which has a better physical meaning than FVC. Besides, 3DFVC has a better applicability than FVC, but it requires the of support Dem. It is when the remote sensing images have a higher spatial resolution that 3DFVC also needs Dem with a higher spatial resolution to meet the demand, which is also its shortcoming.

### 5.2. Advantages for Applying MGWR to Study Spatial Heterogeneity of Vegetation

The spatial distributions of MGWR and GWR are somewhat similar, but they also have some differences. In terms of the models, the biggest difference between MGWR and GWR is that they have different bandwidths (Table 6). The bandwidth of influencing factors is equal to 55 in GWR, and the bandwidth of influencing factors is between 27 and 333 in MGWR. In GWR, it usually uses initial bandwidth and step size to get an optimal bandwidth [53,54]. Therefore, the optimal bandwidth is somewhat affected by initial bandwidth and step size, and only reflects the general level. In addition, it may affect the robustness of GWR and thus indirectly the spatial heterogeneity of GWR. On the contrary, MGWR acquires the bandwidths for multiple variables by means of local adaption. On the one hand, the adaptive process can make MGWR generate multiple bandwidths for explanatory variables. On the other hand, it can also avoid capturing too much noise and bias and make MGWR have a better level of spatial smoothing [55], which is the important reason why MGWR shows a weaker spatial heterogeneity than GWR.

MGWR mitigates the effect of scale noise on spatial heterogeneity, which can allow the spatial distribution of influencing factors to be more compact. It is in both MGWR and GWR that the growth of vegetation is influenced firstly by climate and human activity, secondly by topography and thirdly by soil. Firstly, it is in terms of climate and human activity that the bandwidths of Tem (193 and 55) have a large difference in GWR and MGWR, but the bandwidths of Pre and Ha (43 and 55; 27 and 55) have a small difference in GWR and MGWR. Therefore, it is in Figure 12 that Tem has a large variation in spatial heterogeneity between GWR and MGWR, but Pre and Ha have a small variation. Secondly, in terms of topography, Dem and Slope have the similar phenomenon with Pre, Ha and Tem. Thirdly, in terms of soil, Clay and Silt are not represented spatially due to their lower regression coefficients (Table 3). Furthermore, MGWR not only explores the spatial heterogeneity of vegetation, but it also filters out non-essential influencing factors (Dem, Slope, Clay and Silt). Therefore, it retains the main influencing factors (Pre, Tem and Ha), which can reduce a certain amount of work for the study of vegetation dominant factors.

**Table 6.** Comparison of bandwidth between GWR and MGWR.

| Bandwidth | Pre | Tem | Ha | Dem | Slope | Clay | Silt |
|---|---|---|---|---|---|---|---|
| MGWR | 43 | 193 | 27 | 61 | 333 | 36 | 242 |
| GWR | 55 | 55 | 55 | 55 | 55 | 55 | 55 |

### 5.3. MO's Inspiration for Study on Vegetation Influencing Factors

Since mediating effect model (ME) and moderating effect model (MO) cannot hold simultaneously. Therefore, when MO holds, ME becomes irrelevant. The results show that Pre has a direct effect on vegetation growth and Tem has a moderating effect on vegetation growth (Figure 13), which indicates Pre has different effects on vegetation growth on different temperature conditions. By contrast, previous studies mainly explore the direct relationship between vegetation growth and its influencing factors through the correlation or regression analysis [56,57]. In addition, few studies are to mine the potential relationship between vegetation growth and its influencing factors. It is in terms of Pre that Pre has a direct contribution to vegetation growth, which is generally consistent with the conclusions of previous studies [58,59]. However, it is in terms of Tem that Tem has no direct contribution to vegetation growth but plays a moderating role, which means that Tem is only used as a condition factor for vegetation growth, not as a dominant factor in Northeast China. Therefore, it is tried to explore the intrinsic driving mechanisms of vegetation growth by introducing MO, which can provide an inspiration for the relevant studies in other regions.

### 5.4. Recommendations for Ecological Management of Vegetation

There is a strong zoning characteristic in vegetation. The results of GWR and MGWR show that vegetation in Region I is mainly influenced by human activity, while vegetation in Region II is primarily influenced by climate. Combining with the natural features in Northeast China, here are two suggestions for protecting vegetation. The first suggestion is to adopt a zoning approach to implement ecological management. The specific measures are that vegetation is conserved in Region I via anthropogenic ecological restoration strategies (such as land restoration, afforestation and so on), while vegetation is protected in Region II by natural restoration (such as eco-migration, establishment of nature reserves and so on). The second suggestion is that the relevant policies or regulations should be improved concerning "environmental protection", so as to promote the sustainable development of Northeast China.

### 5.5. Limitations

It is in terms of the influencing factors of vegetation that only those that can be easily quantified are included in this study such as climate, soil, topography and human activity. Besides, there are far more factors affecting the growth of vegetation than those mentioned above [60]. However, some factors (such as grazing, ecological governance or other factors) that are difficult to quantify, are not taken into account in this study and they also may have an impact on vegetation growth. It is in future that a wider range of factors can be considered for inclusion.

## 6. Conclusions

In Northeast China, this study uses 3DFVC to study the spatio-temporal vegetation changes from 2000 to 2021. With the support of GWR and MGWR, the spatial heterogeneity analysis between vegetation and its influencing factors is carried out. In addition, the driving mechanisms on vegetation growth are acquired by the moderating effect model. The main conclusions are reached, as follows:

(**1**) 3DFVC has a better physical meaning than FVC. 3DFVC has a higher regression coefficient and a lower RMSE, which indicates that 3DFVC is better than FVC on vegetation cover extraction. Additionally, 3DFVC has a better applicability than FVC, not only for areas with complex terrain, but also for areas with flat terrain.

(**2**) Vegetation in Northeast China improves overall with a strong zoning characteristic. From 2000 to 2021, vegetation cover shows a fluctuating increasing trend. Spatially, vegetation in Northeast China is dominated by middle high coverage and high coverage with highest vegetation cover in the humid region, second highest vegetation cover in the semi-humid region and lowest vegetation cover in the semi-arid region.

(3)    Vegetation trends are stable in most areas and significant in local areas. 24.36% of vegetation area improves and its spatial distribution is clustered. 15.32% of vegetation area degrades and its spatial distribution is fragmented. The cumulative variance contribution of EOF accounts for 39.7%. $VC_1$ accounts for 25.5%, $EOF_1$ and its time coefficient indicate that vegetation is obviously improved in the semi-humid region with a strong spatial heterogeneity. $EOF_2$ and its time coefficient, $VC_2$ accounting for 14.2%, indicate that vegetation changes sensitively in the semi-arid region with a strong temporal heterogeneity. The mean hurst is less than 0.5, which indicates that vegetation is at some risk of degradation in future. Additionally, it is in future that vegetation changes significantly in the south of Northeast China and continues to be stable in the north of Northeast.

(4)    Vegetation growth is most strongly influenced by climatic and human activity, second most by topography and least by soil. Besides, precipitation plays a leading role on vegetation growth, while temperature and human activity play a moderating role on vegetation growth. What is more, precipitation has a better explanatory power on vegetation growth when temperature is the moderating variable.

**Author Contributions:** Conceptualization, M.L. and Q.Y.; methodology, M.L., Q.Y. and G.L.; validation, M.L., M.Y. and J.L.; formal analysis, M.L. and J.L.; investigation, M.L., G.L., Q.Y. and J.L.; writing—original draft preparation, M.L. and J.L.; writing—review and editing, M.L. and Q.Y.; visualization, M.L. and M.Y.; supervision, Q.Y. and M.Y.; project administration, Q.Y.; funding acquisition, Q.Y. All authors have read and agreed to the published version of the manuscript.

**Funding:** This research was funded by the Third Comprehensive Scientific Investigation Project of Xinjiang Province (No. 2022xjkk1004), the National Natural Science Foundation of China (No. 51874306), the Open Funds of Key Lab for Carbon Neutrality and Territorial Spatial Optimization (No. 2021CNLSO1001), the Fundamental Research Funds for the Central Universities (No. 2021ZDPY0205).

**Acknowledgments:** The authors would thank the editors and the anonymous reviewers for their suggestions. We also appreciate professor Peijun Wang for improving the study.

**Conflicts of Interest:** The authors declare no conflict of interest.

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
