# Peer review of "Spatio-Temporal Changes of Vegetation Cover and Its Influencing Factors in Northeast China from 2000 to 2021"

_remotesensing, doi:10.3390/rs14225720_

Round 1
Reviewer 1 Report
Summary:
This manuscript address on Spatio-temporal changes of vegetation cover and Its Influencing factors in Northeast China from 2000 to 2021, which using moderate resolution imaging spectroradiometer (MODIS) NDVI products. Some interesting results could provide useful information for land use planning, land management, and regional development decision makers. The science and methodology of the manuscript appear sound, and adequately cited. Authors used several models to analyze vegetation cover change from 2000 to 2021, and used a couple models to figure out the mechanisms of vegetation response to climate and human activities. This study is a beneficial exploration of vegetation changes, land use change and environmental changes. One point that could perhaps be strengthened is more of an indication to the readers of the level of uniqueness of the study comparing with previous studies if there is any. I believe a major level of revisions should be made to the paper before it is ready to be considered for publication with Remote Sensing. See the detail comments below.
General Comment:
As I mentioned in summary, it will help readers to better understand the level of uniqueness of the study comparing with previous studies if there is any, add more sentences in the introduction section about what have been done.
Advantages of MODIS data are high temporal frequency with low spatial resolution. Most work based on annual average, what reason to select MODIS, not Landsat?
Some basic concepts need to be clean, such as Land cover, Land use, et al.
How authors deal with multiple spatial resolution data if there are different?
Specific Comments:
Figure 1 need to be modified. First, three panel need to be balance in size, legends: all three legends should be same size and front. Second, title of legend needs to be consistent, I would suggest removing all title of legend. Third, panel b, Land use and land cover is different concept and team, what data do you try to show here (Land use or cover)? Fourth, Panel c: remove title of legend, and add (m or meter) after numbers (-263, 2650)…
Please spell out all Abbreviations at first time use: for example: NDVI …
Table 1: can author add the more information about the datasets, year (period), resolutions, et al.
Reviewer 2 Report
This is an interesting and valuable case study providing important and novel information about 22 years patterns of vegetation cover in Northeast China. Such complete and long-term analysis about this region is new. However, there are additional very interesting novel methodological results about some advanced methods: comparing FVC and 3DFVC and comparing GWR and MGWR. Using other advanced methods enabled the authors analyzing spatial and temporal heterogeneity beside the overall main trends. Other positive aspect is the application of SEM for exploring causal relationships and exploring relative importance of driving factors. The authors used appropriate data for these analyses.
The paper has a logical structure. However, some of the methods were not well described, it was impossible to understand them from the text, and therefore, it was hard to fully understand the related results (see below in my detailed comments). English grammar is an additional problem. Some sentences hard to understand. English needs an overall revision.
Detailed comments
ABSTRACT
Abstract is clearly written. However, all abbreviations (e.g. EOF, Sen, MK, Hurst) should have been defined when they were first mentioned.
L 13 “vegetation changes are relatively stable”. It is not clear how some change is stable (is the first derivative constant?) I suggest writing: “vegetation cover is relatively stable”. Please, check also the numbers you cite here (7.09%) because there are different numbers in the main text.
L 15 hurst index or Hurst index?
L 15-16 “anti-persistence feature” this term was not defined and introduced in methods.
L 18 the word “disturbed” should be deleted from this sentence.
KEYWORDS - some keywords simply repeat the title. These are redundant and could be replaced to other keywords
QUALITY OF FIGURES – Figures are informative and made in good quality. Small problems: 1, dark green and light green colors hard to distinguish for example forests and grasslands in Fig.1b.
2, when black title is written on dark blue color then it is hard to read (e.g. in Fig. 8.b, 8c).
3, “curverd surface” should be “curved surface” in the legends of Fig.3.
INTRODUCTION – It is perfectly written, clear and informative.
MATERIALS AND METHODS – most part is clearly written. However, some parts need a complete revision adding more details and clearer explanations. It was not possible to understand sections: 3.2.2 and 2.2.3. in L 155-166.
L 144 Please, define what is GEE
L 169-170 in equation (12) what is i, j and k ? please define
L 163 references 44 and 45. Here you cited two arbitrary chosen case studies. However, there are many excellent methodological textbooks and reviews about SEM that should be better to cite here.
L 191 was MO correct here? I guess you should write ME here.
RESULTS This section is unusually long and detailed. In general, it was well written, clear and understandable. I enjoyed reading this text and I agree to maintain Results in this long, detailed version. I like Figures 8 to Figures 12 which are novel, very interesting results. Unfortunately some methods were not well explained in Method section and therefore the related results (Figures 10 and 11) are not clear. I found some other parts in Results section that need clarification:
L 238-245 and Table 2 – not clear, however the previous text in my opinion already fully explained the advantage of 3DFVC. Therefore I suggest deleting the text L 238-245 and Table 2.
L 258 “changes tend to be stable” – this is not clear, not logical.
L 274-285 There are many confusing terms here. Please, correct and clarify: stable changes, stable trend, significant trend (what was the statistical test?). The numbers reported here do not fit to the number written in Abstract.
Fig.9. I could not find the definitions for the 5 levels of degradation.
L 293-315 This text is not clear, it needs a complete revision (together with the related Methods section).
L 319-329 It is not clear how the prediction for future was made. In general such prediction is always based on some assumptions. I miss the related background and explanations. What is the time scale for the predictions on Fig.11. ?
L 396 I think “Pre” should be replaced by “Ha”
DISCUSSION is very short, poorly written. I completely missed the comparisons with other studies and the related citations . There are some interesting problems that could be discussed in Discussion section:
1, Comparing GWR and MGWR how the band widths were selected (cf Table 4)? Why the results with MGWR were less heterogeneous (Fig 12)?
2, Why only moderating effects were evaluated in Fig 13? Why mediating, indirect effects were not important? Why overall SEM results were presented from complex spatiotemporal processes with several dimensions of heterogeneity documented here in other results (Figs 8-12)?
CONCLUSIONS – It was OK, maybe too long and it also had the same problems reporting the illogical “stable changes”
Round 2
Reviewer 1 Report
The authors did a decent job in revising the manuscript. They have addressed my main concerns and taken care of minor comments/editorial changes. I recommend that the manuscript be accepted for publication.
Reviewer 2 Report
The authors made a very careful revision considering all points asked by the Referee. I am completely satisfied with the changes they made. They gave very detailed and clear responses, documenting very carefully all changes and improvements.
During my reading of the revised manuscript, I found only a small typo
L170
“Table 2. The ategories for vegetation cover trend.”
Should be Table 2. The categories for vegetation cover trend.
This small error can be amended during proof reading.